# Mitochondria-Targeted Fluorescent Nanoparticles with Large Stokes Shift for Long-Term BioImaging

**DOI:** 10.3390/molecules28093962

**Published:** 2023-05-08

**Authors:** Xiao Li, Tao Zhang, Xuebo Diao, Yu Li, Yue Su, Jiapei Yang, Zibo Shang, Shuai Liu, Jia Zhou, Guolin Li, Huirong Chi

**Affiliations:** 1Key Laboratory of Microecology-Immune Regulatory Network and Related Diseases, School of Basic Medicine, Jiamusi University, Jiamusi 154000, China; 2Department of Obstetrics and Gynecology, The Second Affiliated Hospital of Harbin Medical University, 246 Xuefu Road, Nangang District, Harbin 150081, China; 3Key Laboratory of Systems Biomedicine (Ministry of Education), Shanghai Center for Systems Biomedicine, Shanghai Jiao Tong University, 800 Dongchuan Road, Shanghai 200240, China; 4School of Chemistry and Chemical Engineering, Frontiers Science Center for Transformative Molecules, Shanghai Jiao Tong University, 800 Dongchuan Road, Shanghai 200240, China; 5Faculty of Science, University of British Columbia, 2329 West Mall, Vancouver, BC V6T 1Z4, Canada; 6Department of Radiology, Shanghai Sixth People’s Hospital Affiliated to Shanghai Jiao Tong University School of Medicine, Shanghai Jiao Tong University, Shanghai 200233, China; 7Department of Stomatology, Shanghai Eighth Peoples Hospital, 8 Caobao Road, Shanghai 200000, China

**Keywords:** AIE-active fluorescent probe, MITO targeted, nanoparticle, real-time, long-term imaging

## Abstract

Mitochondria (MITO) play a significant role in various physiological processes and are a key organelle associated with different human diseases including cancer, diabetes mellitus, atherosclerosis, Alzheimer’s disease, etc. Thus, detecting the activity of MITO in real time is becoming more and more important. Herein, a novel class of amphiphilic aggregation-induced emission (AIE) active probe fluorescence (AC-QC nanoparticles) based on a quinoxalinone scaffold was developed for imaging MITO. AC-QC nanoparticles possess an excellent ability to monitor MITO in real-time. This probe demonstrated the following advantages: (1) lower cytotoxicity; (2) superior photostability; and (3) good performance in long-term imaging in vitro. Each result of these indicates that self-assembled AC-QC nanoparticles can be used as effective and promising MITO-targeted fluorescent probes.

## 1. Introduction

Mitochondria, as the powerhouses inside the cells, play a significant role in various physiological processes such as cell information transmission, cell differentiation, and apoptosis [1,2,3,4]. Mitochondria are also key regulators of the host response to viral infection, inflammation, and immunity, and this is critically exemplified during infection by severe acute respiratory syndrome coronavirus 2 (SARS-CoV-2). Furthermore, MITO is a key organelle related to different human diseases including cancer, diabetes mellitus, atherosclerosis [5], Alzheimer’s disease [6], etc. The number and distribution of mitochondria in cells have been shown to be closely related to the process of cancer cell metastasis and normal cell apoptosis.

Therefore, it is urgent to engineer a method that is suitable for visualizing MITO. Various imaging methods have evolved, such as fluorescence visualization, Raman scattering imaging, magnetic resonance imaging, photoacoustic imaging, ultrasound imaging, X-ray radiography, and positron emission tomography. Among them, fluorescence methods are widely used by virtue of their simplicity, non-invasiveness, great sensitization, and high spatiotemporal resolution [7,8,9,10,11]. So far, different sorts of fluorescent probes have been evolved for visualizing MITO [12,13,14,15,16,17,18,19,20,21,22], such as Rhodamine 123, MitoTrackers, MitoTrackers Red CMXRos, Green FMs, and MitoTrackers Orange CMTMRos. However, conventional fluorescent probe applications have been restricted by non-specific binding, photobleaching, poor photostability, cytotoxicity, spectral over-lap with bio substrate autofluorescence, and inherent fluorescence quenching (under the circumstance of aggregation in the aqueous medium). Some fluorescent probes may be toxic to cells at high concentrations, leading to cell death and inaccurate results. Many fluorescent probes can bind to cellular components other than mitochondria, leading to non-specific staining and inaccurate results. Fluorescent probes can be susceptible to photobleaching, a process where the fluorescence signal decreases over time due to prolonged exposure to light. This can limit the duration of imaging and make it difficult to obtain accurate results. Photobleaching can also result in a decreased signal-to-noise ratio, making it challenging to distinguish the fluorescent signal from the background noise. Even though more fluorogenic substances transformed into a synthetic polymer microsphere via an elaborate control, the aggregation-caused quenching (ACQ) issue hampers the manufacture of polymer beads with highly emissive features. Further for an ideal fluorescent probe, a large Stokes shift (typically over 80 nm) is generally favorable to minimize the cross-talk between the excitation source and fluorescence emission [23,24]. Nevertheless, typical organic dyes, such as boron-dipyrromethene or phthalocyanine, tend to perform undesirable background interference exhibits, because of small Stokes shifts (≈7 to 20 nm), which partially reabsorb emitted photons [25]. Although an energy transfer strategy is probably introduced between these fluorophores to enlarge the gap between excitation and emission wavelengths, the entire system tends to be structurally complicated, coupled with necessary laborious synthetic exerts [26,27,28]. In contrast to conventional fluorescent dyes with the fluorescence of ACQ, aggregation-induced emission (AIE) fluorophores are capable to work as an ideal “turn-on” fluorescent probe for bioanalysis, since they exhibit shining fluorescence in the aggregated state while very weak fluorescence in a good solvent [29,30,31,32,33,34,35,36,37,38,39]. Admittedly, AIE luminous materials partially address the issue that existed in conventional dyes, paving the way for a new AIE fluorescent probe development with optimistic characteristics for bioimaging. However, a limited application is observed in tracking biological processes and pathological pathways over long periods. This lies in the fact that most currently existing AIE probes exhibit small Stokes shifts and lack specific organelle-targeting capacity. Moreover, their application in tracking biological processes and pathological pathways over a long period of time is limited.

To meet these challenges, we constructed an AIE-active fluorescent probe (QC) with superior photostability and large Stokes shift, which can be employed for monitoring vesicular transportation and long-term noninvasive imaging. This high-performance AIE-active fluorescent probe was designed and synthesized using a quinoxalinone scaffold as the chromophore core. In contrast to previous AIE-active fluorescent probes reported [33,40,41,42,43,44,45,46,47,48,49], quinoxalinone-based AIE-active fluorescent probes possess a long emission wavelength because of the broad absorption wavelength of the quinoxalinone scaffold. Afterward, AIE-active QC was then conjugated to the mitochondria-targeted peptides [50] Ac-Lys- (D-Arg)- (Cha)- (D-Arg)- (Cha)- (D-Arg)- (Cha)- (D-Arg) (AC) to enhance solubility and transmit the capacity of a targeting organelle (Figure 1), resulting in a conjugate derivative named AC-QC. Owing to the amphiphilic feature of the AC-QC conjugate, when AC-QC conjugate was dispersed in an aqueous solution, the advent of self-assembly was recognized as the formation of nanoparticles with hydrophilic peptide groups topping at the surfaces. The hydrophobic AIE components were simultaneously aggregated into the cores, leading to the high dispersibility of these obtained AC-QC nanoparticles in an aqueous environment. The surface peptide could become MITO-targeted, and the AIE molecule inside the nanoparticle could achieve noninvasive long-term imaging under light illumination.

## 2. Results

### 2.1. Development and Depiction of AIE Dye (QC)

QC chemical formula was validated by ^1^H NMR, ^13^C NMR (Appendix A), and FTIR measurements (Figure 2A). As shown in Figure 2A, the peaks at 1657 cm^−1^ referred to the stretching vibration of C=O; the peaks at 1621 and 1604 cm^−1^ could be assigned to the stretching vibration of C=C; and the peaks at 1575 and 1535 cm^−1^ were attained through the stretching vibration of C=N. QC excitation and emission spectra in different organic solvents were estimated with peaks centered at about 410 and 520 nm, respectively (Figure 2B). QC excitation and emission spectra in PBS solvents were estimated with peaks centered at about 394 and 524 nm (Figure 2D). The peaks of excitation and emission spectra did not change in different solvents. Subsequently, the AIE was characterized in mixed solvents of Gly/MeOH with MeOH fraction ranging from 0% to 90%. Specifically, when the methyl alcohol concentration elevated, the intensity of absorption of QC significantly declined because of its hydrophobic characteristics and the production of aggregates. Notably, the fluorescence intensities of QC were markedly improved at a simultaneous formation of aggregates, exhibiting green in an AIE-active characteristic (Figure 2C). Thereupon, we inferred that QC may serve as a potential fluorescent probe for bioimaging based on these optical properties.

Figure 2E depicts the absorption peak of the UV–vis spectrum of QC in DMSO. The 425 nm absorption band is the result caused by a p–p* transition of the conjugated backbone. In comparison, the broad absorption from 280 to 450 nm originates through charge transfer between the fluorene (donor) units and the thiadiazoloquinoxaline.

### 2.2. Development and Depiction of AC-QC Nanoparticles

Given the amphiphilic feature of the AC-QC conjugate, self-assembly occurred to form nanoparticles (with a cover of hydrophilic peptide groups at the surfaces) when AC-QC conjugate was dispersed in an aqueous solution. Furthermore, the hydrophobic AIE components were aggregated into the cores, leading to the high dispersibility of these obtained AC-QC nanoparticles in an aqueous environment. The morphology and size of AC-QC nanoparticles were specified by dynamic light scattering (DLS) and transmission electron microscopy (TEM). According to Figure 3B, the distribution size histogram demonstrates a unimodal distribution, and the mean hydrodynamic diameter is 122.4 nm with a PDI of 0.247. Some spherical nanoparticles could be clearly recognizable, which indicated the AC-QC nanoparticles from the self-assembly of the AC-QC conjugate. The TEM average diameter of the AC-QC nanoparticles as defined utilizing Image J is 110 nm (Figure 3A and Appendix A) [51], identical to that indicated by DLS measurement. It was noteworthy that the particle size characterized by TEM was to some extent smaller compared to the size distribution in PBS, which might be attributed to the drying-induced shrinkage of the self-assembly process. The molecular weight of the AC-QC conjugate was determined through mass spectrometry (MS). MS: *m*/*z* (ESI) calculated for C_84_H_130_N_24_O_14_ [M-H], 1700.08, found 1699.8 (Appendix A). AC-QC excitation and emission spectra in DMSO solvents were estimated with peaks centered at about 407 and 488 nm. AC-QC excitation and emission spectra in PBS solvents were estimated with peaks centered at about 394 and 524 nm (Figure 3C). The absorption peak of the UV–vis spectrum of AC-QC in DMSO is shown in Appendix A.

### 2.3. In Vitro Cell Toxicity

The nanoparticle cytotoxicity of AC-QC against IOSE-80 and L929 cells was evaluated by using the standard MTT assay (Figure 4 and Appendix A). The concentration impact of AC-QC nanoparticles on the overgrowth of IOSE-80 and L929 cells is indicated in Figure 4. The AC-QC nanoparticle showed low cell cytotoxicity at both low (10 μg/mL) concentration and high concentration (100 μg/mL). The MTT outcome showed that the IOSE-80 and L929 remain higher than 80% at the maximum micelle concentration (100 μg/mL). All results showed that AC-QC nanoparticle was noncytotoxic to living cells under our cell-imaging conditions. Since AC-QC nanoparticles reveal lower cell toxicity to the IOSE-80 and L929 cells, they can be used as a fluorescent probe for bioanalysis.

### 2.4. Colocalization Study

Based on the existing self-assembled AC-QC nanoparticles, confocal imaging-based colocalization investigations were conducted by including the cells with AC-QC nanoparticles and different organelle trackers. By this means, the location of nanoparticles inside the cells was supposed to be identified. At the beginning stage, the location of AC-QC nanoparticles in cells was investigated by a MITO tracker. L929 cells were consequently preserved with AC-QC nanoparticles (10 μg/mL) for 4 h. Then, the MITO tracker (1 × 10^−6^ M) was added to the culture media, allowing a 30 min incubation. At this stage, the fluorescence (green) of the AC-QC nanoparticles overwhelmed that of the MITO tracker (red fluorescence), with a Pearson correlation coefficient (PC) value of up to 0.9011 ± 0.05 (Figure 5A). On the contrary, the nanoparticle-based probe could not label the endoplasmic reticulum (ER), Golgi apparatus (GOLI), lysosome (LYSO), or nuclei, with PC values of 0.2184 ± 0.03, 0.2934 ± 0.05, 0.2526 ± 0.02, and 0.2127 ± 0.02, respectively, showing a more desirable capacity of AC-QC for MITO staining (Figure 5). In addition, AC-QC nanoparticles also exhibited the same MITO-targeting capacity for other cell lines, such as IOSE-80 cells (Figure 5B). These results validated our hypothesis that AC-QC nanoparticles were potentially an excellent probe to monitor MITO in real time. Moreover, this could indicate that the living cell membrane is permeable to AC-QC nanoparticles.

### 2.5. Photostability of AC-QC Nanoparticles

The light stability of probes is one of the most important indexes to evaluate the performance of the probes. To further validate the potential of AC-QC nanoparticles to serve as a stable imaging agent in a biological environment, AC-QC nanoparticles’ photostability was assessed under different conditions. First, we investigated its photostability by comparing the MITO staining of IOSE-80 cells, at which stage AC-QC nanoparticles (bottom) and a commercially available MITO tracker (upper) were used under continuous light irradiation for 20 min. It could be assumed that a great fluorescence signal in terms of the AC-QC nanoparticle probe (10 μg/mL) could be detected with insignificant signal decay through the irradiation (Figure 6). In contrast, the fluorescence intensity of the MITO tracker reduced at a noticeable rate and became insignificant after 10 min irradiation owing to the photobleaching (Figure 6). The good photostability could support much more working time for the AC-QC nanoparticles under a laser source. Because of the superior photostability of AC-QC nanoparticles, we then explored their in vitro long-term imaging capacity. IOSE-80 cells were initially incubated with AC-QC nanoparticles (10 μg/mL) at 37 °C for 4 h (Labeled as Day 0, generation one). Then, the treated cells were subcultured by specified time intervals (24 h) and the fluorescence signals were imaged (Day 3, generations one to four). In the following stage, the culture medium was removed for each cell passage by rinsing the cells twice with ice-cold PBS. In the preliminary phase (Labeled as Day 0, generation one), the AC-QC nanoparticles’ bright green fluorescence signal was observed (Figure 7). With an increased incubation time (Day 1 to 3, generation two to three, respectively), green fluorescence reduces progressively corresponding to cell proliferation. The AC-QC nanoparticles were divided into daughter cells. On the other hand, the green signal was still observable after 3 d (the third generation) (Figure 7), appearing as a striking contrast to the MITO tracker, which showed that the AC-QC nanoparticles could function as a fluorescent probe for long-term tracing and imaging of cells. Moreover, the absorbance and fluorescence of AC-QC nanoparticles and the MITO tracker upon the irradiation were quantified (Appendix A).AC-QC nanoparticles are highly photostable and exhibit a much higher photostability than the commercialized MitoTracker, which is critical for building smarter fluorescent probes to ensure long time imaging.

## 3. Discussion

MITO is a key biomarker related to different human diseases of cancer, diabetes mellitus, atherosclerosis, Alzheimer’s disease, etc. Hence, detecting the activity of MITO in real time is becoming more and more important. As is presented, the fluorescence intensities of QC were enhanced at a considerable rate when aggregates formed, with an exhibition of green in an AIE-active manner. Subsequently, we successfully synthesized an AC-QC conjugate. Owing to the amphiphilic feature of the AC-QC conjugate, when the AC-QC conjugate was dispersed in an aqueous solution, self-assembly occurred to form nanoparticles with hydrophilic peptide groups covered at the surfaces, while the hydrophobic AIE components were aggregated into the cores, leading to MITO being targeted. It can be thusly inferred that AC-QC may serve as an ideal fluorescent probe for bioimaging based on these optical properties. The MTT result indicates that the IOSE-80 and L929 cell viability is still higher than 80%, even at the highest micelle concentration (100 μg/mL). Since AC-QC nanoparticles reveal low cytotoxicity to the IOSE and L929 cells, they can be employed as a fluorescent probe for bioanalysis. The colocalization study indicates that AC-QC has good MITO staining capacity. These results validated our hypothesis that AC-QC nanoparticles could act as an excellent probe to monitor MITO in real time. For the AC-QC nanoparticle probe (10 μg/mL), strong fluorescence signals could be observed with almost no signal decay under continuous light irradiation for 20 min. The result shows that the nanoparticles present an ideal photobleaching resistance, as the green signal can still be observed after 3 d (the fourth generation), appearing as a stark contrast to the MITO tracker, which indicates that the AC-QC nanoparticles can act as a fluorescent probe for long-term cellular tracing and imaging. All of these results demonstrate that the self-assembled AC-QC nanoparticles can be used as effective and promising MITO-targeted fluorescent probes.

## 4. Materials and Methods

### 4.1. Development of AIE Dye (QC)

The QC synthetic route was demonstrated in Figure 1.

The 4-methoxy-o-Phenylenediamine (0.1 mol, 10.8 g), suspended in 150 mL of anhydrous ethanol, formed a combination, which was cooled in an ice bath sequentially. Afterward, a solution of ethyl pyruvate (0.12 mol, 13.92 g) in 10 mL of anhydrous ethanol was added by dropping over 20 min under stirring. The reaction of the resulting solution lasted for 12 h under room temperature, allowing precipitate to form. After filtering, the precipitate was washed via ethanol, and dried through the vacuum to give the product 7-methoxy-3-methylquinoxaline-2 (1H)-one (compound **3**) (13.6 g, yield: 85%) as a yellow solid without purification. ^1^H NMR (400 MHz, CDCl_3_): δ = 12.18 (s, 1H), 7.57 (d, *J* = 8.0 Hz, 1H), 6.83–6.86 (m, 1H), 6.70–6.72 (m, 1H), 3.78 (s, 3H), 2.32 (s, 3H) ppm; ESI-MS *m*/*z*: calculated for C_10_H_10_N_2_O_2_ [M + 1]^+^ 190.2, found 190.2.

Compound **3** (52.6 mmol, 10 g) and K_2_CO_3_ (78.9 mmol, 10.9 g) were immersed in acetone (100 mL), and a 3-bromoprop-1-ene (63.1 mmol, 7.6 g) solution was supplemented drop-wise. The combination was oil-bathed at 62 °C with stirring for 12 h. The obtained solvent was extracted by use of a rotary evaporator, and the leftover material was separated between distilled water (20 mL) and ethyl acetate (EA) (40 mL). The organic layer was separated and dried over anhydrous MgSO_4_, filtered, and concentrated in a rotary evaporator to obtain a crude by-product, which underwent a filter through chromatography with silica gel (hexane: EA = 20:1, *v*:*v*), to afford 10.1g of 5 as a yellow solid, yielding 83.4%.^1^H NMR (400 MHz, CDCl_3_): δ = 7.71 (d, *J* = 8.0 Hz, 1H), 6.90 (d, *J* = 8.0 Hz, 1H), 6.70 (s, 1H), 5.99–5.85 (m, 1H), 5.28 (d, *J* = 12 Hz, 1H), 5.18 (d, *J* = 12 Hz, 1H), 4.85–4.88 (m, 2H), 3.88 (s, 3H), 2.55 (s, 3H) ppm; HRMS: *m*/*z* (ESI) computed for C_13_H_12_N_2_O_2_ [M + 1]^+^ 231.1055, retrieved 231.1118.

A solution of compound **5** (6.51 mmol, 1.5 g) in acetic acid (15 mL) was supplemented with N- (4-formyl phenyl) acetamide (3 mmol, 1.28 g) and concentrated sulfuric acid as a catalyst. The acquired solution was warmed up to 50 °C and allowed 8 h for reaction; thereafter, a mixture as output was concentrated in a rotary evaporator, with water (20 mL) and ethyl acetate (50 mL) supplemented in the process. Aside from collecting the organic layer, the watery phase was basified with K_2_CO_3_ and attained with ethyl acetate, and the dehydration of the combined organic phase was performed using anhydrous MgSO_4,_ and filtration was performed by silica chromatography (Di-chloromethane (DCM): Methanol = from 20:1 to 10:1) to yield 1.51 g of 7 as a red solid, yield: 61.6%. ^1^H NMR (400 MHz, CDCl_3_): δ = 8.45–8.42 (m, 1H), 7.74–7.67 (m, 2H), 7.60–7.36 (m, 3H), 7.35–7.25 (m, 2 H), 7.11–6.95 (m, 1H), 6.05–5.94 (m, 1H), 5.30 (d, *J* = 12.0 Hz, 1H), 5.21 (d, *J* = 12.0 Hz, 1H), 4.85–4.98 (m, 2H), 3.91 (s, 3H), 2.16 (s, 3H) ppm; HRMS: *m*/*z* (ESI) computed for C_22_H_22_N_3_O_3_ [M + 1]^+^ 376.1661, discovered 376.1658.

A solution of compound **7** (2.66 mmol, 1 g, 1.0 eq) in Dichloromethane (15 mL) at −40 °C was added drop-wise 3 equivalents of 1M BBr_3 (_7.98 mmol, 8 mL, 3.0 eq) to dichloromethane. Sequentially, the combination was blended at room temperature for 12 h. Upon the reaction accomplishment, as determined through thin-layer chromatography (TLC), flowed into ice, and the aqueous portion was attained with EtOAc and dried. It was further purified by column chromatography eluted with (Dichloromethane (DCM): Methanol = from 100:1 to 20:1, *v*:*v*) to obtain 0.49 g of compound **8** as a red solid, yield: 51.0%. ^1^H NMR (400 MHz, CD_3_OD): δ = 8.72–8.69 (m, 1H), 7.77–7.72 (m, 5H), 7.59–7.56 (m, 2H), 7.31–7.26 (m, 1 H), 6.05–5.96 (m, 1H), 5.40 (d, *J* = 12.0 Hz, 1H), 5.31 (d, *J* = 12.0 Hz, 1H), 5.23–5.12 (m, 2H), 2.17 (s, 3H) ppm; HRMS: *m*/*z* (ESI) calculated for C_21_H_20_N_3_O_3_ [M + 1]^+^, 362.1505, found 362.1501

Compound **8** (1.0 mmol, 375 mg) and K_2_CO_3 (_1.5 mmol, 215 mg) were suspended in acetone (20 mL). Then, (E)-4-bromobut-2-enoic acid (1.2 mmol, 207 mg) solution was supplemented drop-wise into the combination, which was stirred in the process of an oil bath at 62 °C for 12 h. After the reaction, the solvent was removed by use of rotary evaporator, and the leftover was partitioned between distilled water (20 mL) and ethyl acetate (EA) (40 mL); thereafter, the organic layer was divided and dehydrated through anhydrous MgSO_4_, and a crude product was filtered, concentrated using a rotary evaporator, and then purified with silica gel chromatography ((Dichloromethane (DCM): Methanol = from 100:1 to 20:1, *v*:*v*)) to afford 110.6 mg of the compound **9** as a red solid, yield: 35%.^1^H NMR (400 MHz, CD_3_OD): δ = 8.04–8.01 (m, 1H), 7.73–7.66 (m, 2H), 7.64–7.61 (m, 3H), 7.48–7.44 (m, 1 H), 7.24–7.22 (m, 1H), 7.14–7.08 (m, 1H), 6.09–5.99 (m, 1H), 5.49 (s, 1H), 5.38–5.31 (m, 1H), 5.25–5.19 (d, *J* = 12.0 Hz, 1H), 5.14–5.07 (m, 1H), 4.03–3.94 (m, 2H), 3.35 (s, 3H), 2.15 (s, 3H) ppm; HRMS: *m*/*z* (ESI) calculated for C_26_H_26_N_3_O_5_ [M + 1]^+^, 460.1872, found 460.1863 (Appendix A).

Compound **9** (0.218 mmol, 100 mg, 1.0eq) was immersed in methanol (5 mL). Then, a sodium hydroxide aqueous solution (0.4 mL, 1mol/L in water, 2.0eq) was supplemented drop-wise in the combination, which was blended at 25 °C for 12 h. The solvent was extracted through rotary evaporation after completing the reaction, and the leftover was partitioned among 10 mL of distilled water and ethyl acetate (EA) (10 mL). The organic layer was divided, dehydrated with anhydrous MgSO_4_, filtered, and concentrated using a rotary evaporator to retrieve a crude product that was then exposed to purification using silica gel chromatography (Dichloromethane (DCM): Methanol = from 80:1 to 10:1, *v*:*v*) to afford 63 mg of the compound **10** (QC) as a red solid, yield 65%. ^1^H NMR (400 MHz, DMSO): δ = 9.32 (m, 1H), 7.70–7.60 (m, 1H), 7.51–7.42 (m, 1H), 7.39–7.21 (m, 3 H), 6.72–6.42 (m, 3H), 6.36–6.19 (m, 1H), 5.68–5.55 (m, 1H), 5.04–4.78 (m, 3H), 4.63–4.49 (m, 2H), 2.99–2.89 (m, 2H), 2.26 (s, 3H). HRMS: *m*/*z* (ESI) computed for C_25_H_24_N_3_O_5_ [M + 1]^+^ 446.4830, retrieved 446.1709 (Appendix A).

### 4.2. Production of AC-QC Nanoparticles

The synthetic pathway of AC-QC nanoparticles was demonstrated in Figure 2.

To prepare AC-QC, QC (compound **10**) and AC (compound **11**) were conjugated together in a DMSO/water cosolvent, as described below. (The ratio of DMSO to distilled water in the cosolvent was 1:9). AC powder (6.4 mg) was dissolved in phosphate-buffered saline (PBS) (pH = 7.4, 9 mL). QC (2.23 mg) was dissolved in DMSO (1 mL), followed by the addition of the equivalent of NHS (0.57 mg) and EDC (0.95 mg) about QC. The AC and QC solutions were allowed to dissolve completely, and the reaction mixture was gently stirred for 1 day. The reaction-obtained mixture was dialyzed for 3 days (MWCO =1.0 KDa) against methanol and distilled water to remove unconjugated QC and DMSO. Furthermore, throughout the process, the water was replaced every 4 h. The crude product was further separated via the preparative liquid chromatography and then lyophilized to produce 6.37 mg of compound **12** (AC-QC) as a red solid powder, yielding 73.8%.

### 4.3. Absorption Spectra

At room temperature (RT), the absorption spectra were estimated with a UV–visible spectrophotometer at 200–800 nm. The probe QC (1 mL, 0.01 mM) was supplemented to 1.5 mL cuvettes, and their respective absorption spectroscopy was estimated in DMSO solution.

### 4.4. Fluorescence Analysis

The stock of fluorescent dye QC used for fluorescence spectral analysis was prepared in DMSO, THF, and Gly. The preparation was diluted to 0.01 mM in DMSO, THF, and Gly solution and was separately added to the 4 mL cuvettes, and the fluorescence intensity of the mixture was measured at λex = 410 nm.

After production, its optical features were described in mixed solutions with different ratios of glycerin/methyl alcohol (Gly/MeOH).

### 4.5. Cell Culture and Internalization

L929 cells (mouse fibroblast cell line) and IOSE-80 cells (human ovarian cell line) were purchased from ATCC (American Type Culture Collection). Initially, L929 cells and IOSE-80 cells were cultured in a culture dish, on the condition of 37 °C with 5% CO_2_ in Dulbecco’s modified Eagle medium (DMEM), 10% fetal bovine serum (FBS), and 1% antibiotics (100 units per mL streptomycin and 100 units per mL penicillin). After culturing for 12 h, DMEM was removed, and three times as many of the cells were rinsed with phosphate-buffered saline (PBS) buffer.

### 4.6. Cytotoxicity Measurements

An MTT (3-(4,5)-dimethylthiahiazo (-2-yl)-3,5-diphenytetrazolium bromide) viability assay against IOSE-80 and L929 cells was performed to determine the cytotoxicity of AC-QC nanoparticles. In 96-well plates, the cells were cultured with a density of 10^4^ cells in 200 μL DMEM in each well. After 12 h, 10 μL of PBS solution of AC-QC nanoparticles with different concentrations were supplemented to the wells, and were allowed another 24 h or 48 h to culture. A 20 μL MTT assay stock solution of 5 mg mL^−1^ was added afterward. Sequentially, the solution was removed and 200 μL DMSO was added after 4 h. After 10 min vibrating, a BioTek Synergy H4 instrument was used for estimating the wavelength absorbance of the solution at 490 nm.

### 4.7. Confocal Imaging of QC and AC-QC in Living Cells

The IOSE-80 and L929 cells with good cell viability were inoculated in AC-QC nanoparticles (10 μg/mL) at 37 °C for 4 h and then washed thrice via PBS buffer at pH 7.4. The fluorescent images were obtained via a microscope with a confocal fluorescence feature (Leica SP8) utilizing an objective lens (×20). The excitation wavelength was 405 nm when the cells were incubated with AC-QC nanoparticles, while 503 nm when the incubation of cells was performed with various organelle trackers. The fluorescent images were analyzed using Image J [51,52].

## 5. Conclusions

As has been mentioned, QC, a novel AIE fluorescent dye, was designed and synthesized. The quinoxalinone’s fluorophore is conducive to the dye for presenting excellent properties in chemical stability, superior photostability, and large Stokes shift. Then, MITO-targeting AC-QC nanoparticles were designed and synthesized. Due to the amphiphilic feature of the AC-QC conjugate, when dispersed in an aqueous solution, it self-assembles to form nanoparticles coated with hydrophilic peptide groups, while hydrophobic AIE components aggregate into the core, achieving MITO targeting of AC-QC nanoparticles in aqueous environments. AC-QC nanoparticles have a good performance of MITO targeting and can visualize the mitochondria in viable cells with good biocompatibility. Meanwhile, strong fluorescence signals could be observed during long-time irradiation of AC-QC nanoparticles, showing the strong photostability and the good ability of long-term imaging in vitro. These results confirm that AC-QC nanoparticles are a better mitochondrial probe for confocal microscopic imaging of living cells than the commercial dye MitoTracker. These findings suggest that AC-QC nanoparticles can function as a potent tool to observe MITO in biological samples and show great essential application in the biomedical branch.

## Data Availability

The data presented in this study are available on reasonable request from the corresponding author.

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
