# Peer review of "Mitochondria-Targeted Fluorescent Nanoparticles with Large Stokes Shift for Long-Term BioImaging"

_molecules, 2023, doi:10.3390/molecules28093962_

Round 1
Reviewer 1 Report (Previous Reviewer 1)
This revised version of the manuscript "Mitochondria–Targeted Fluorescent Nanoparticles with Large Stokes Shift for Long-term BioImaging " by Li Xiao et al. has been significantly improved. Although the authors have not responded adequately to question 1 and 2, the corrections made in the manuscript have significantly improved its scientific soundness. There are very few corrections that should be carried out (mostly typing errors), and the recommendation is to accept the manuscript for publication.
Please provide a reference for ImageJ and the identical DLS and TEM radius may be changed to comaprable.
Author Response
Please see the attachment.

Reviewer 2 Report (Previous Reviewer 2)
Regarding the revised version of the manuscript “Mitochondria–Targeted Fluorescent Nanoparticles with Large Stokes Shift for Real-Time Imaging” by Huirong et al. some issues must be clarified:
- The manuscript presents now a section “Results and Discussion” and a second one “Discussion”. I could observe that the authors in general changed the title of these sections. I strongly encourage them to merge both sections into only one section named “Results and Discussion” as almost 100% of papers dealing with chemistry do.
- Despite Scheme 2 presenting now numbers for the chemical structures, please note that this scheme lacks quality since the numbers are way bigger than any other number in this scheme. Please revise.
1. What is the main question addressed by the research? Optical sensors based on nanoparticles for bioimaging 2. Do you consider the topic original or relevant in the field? Does it address a specific gap in the field? Yes. This manuscript does not address any specific gap in the fields since the literature reports several NPs for the same application. 3. What does it add to the subject area compared with other published material? New materials for sensing are relevant. 4. What specific improvements should the authors consider regarding the methodology? What further controls should be considered? Regarding the methodology, the manuscript is ok 5. Are the conclusions consistent with the evidence and arguments presented and do they address the main question posed? Yes. 6. Are the references appropriate? Yes. 7. Please include any additional comments on the tables and figures. My major concern is about the organization of the manuscript and the way that the topics are presented. My report: "The manuscript presents now a section “Results and Discussion” and a second one “Discussion”. I could observe that the authors in general changed the title of these sections. I strongly encourage them to merge both sections into only one section named “Results and Discussion” as almost 100% of papers dealing with chemistry do"
Author Response
Please see the attachment

Reviewer 3 Report (New Reviewer)
In this manu, an AIE-active fluorescent probe (QC) with superior photostability and large Stokes shift was constructed for monitoring vesicular transportation and long-term noninvasive imaging. And AIE-active QC was then conjugated to the mitochondria targeted peptides to achieve noninvasive long-term imaging for mitochondria under light illumination.
Major points:
1. The fluorescent and UV-Vis spectrum of AC-QC nanoparticles need to be tested.
2. In Fig5, The fluorescence of AC-QC nanoparticles almost completely overlaid with that of the MITO tracker. However, AC-QC nanoparticles still overlapped with other trackers (ER, Golgi and LYSO) to some extent. That means the specificity of AC-QC nanoparticles targeting MITO is not so good. Please give some reasonable explanations or design another experiment to prove the specificity of AC-QC nanoparticles targeting MITO.
3. In Fig7, the long-term cellular tracking was from day 0 to day3. However, the cytotoxicity data were only for 48h. So, it suggests that cell viability for 72h should be tested.
Minor points
1. In Fig3, please provide the size distribution of TEM and the intensity data of DLS.
2. In the Lin 148-149, there were some grammar mistakes.
Author Response
Please see the attachment

Reviewer 4 Report (New Reviewer)
The paper presents an attempt to synthesize a new mitochondria staining fluorescent dye based on synthesized nanoparticles providing aggregation-induced emission. In general the idea is interesting and some of the results obtained are promising. However at the current stage the paper cannot be recommended for publication since a considerable part of the results is not convincing and more experiments need to be performed, that definitely will require relatively long time. Therefore I recommend to reject the paper and reconsider it when authors would perform the necessary additional experiments. My major concerns are:
1. The fluorescent properties of the QC dye were tested in methanol, glycerol, DMSO and THF. None of those is a biological medium. Similar data is needed for water or PBS or DMEM.
2. The fluorescent properties of the dye could change dramatically in the AC-QC conjugate. Therefore similar experiments should be performed for the nanoparticles in water or PBS or DMEM.
3. The colocalization images are of far too low resolution and are not convincing. Experiments with much higher resolution are required.
Besides that there synthesized nanoparticles are quite big, their size is even comparable with the size of mitochondria and lysosomes. How they are supposed to penetrate in these organelles?
4. The data on comparison of photostability is also not convincing since no info is given on power density of the applied radiation.
5. The statement in Conclusions that the dye eliminates spectral overlap with autofluorescence is wrong. According to the spectra shown in Fig. 2 it fluoresces exactly in the same spectral band as the cellular autofluorescent species. And it is a considerable drawback.
Besides that
- authors need to extend the introduction section and indicate in detail the drawbacks of all common fluorescent probes used to stain mitochondria.
- in the title of sec. 2.1 authors call the QC dye a photosensitizer. What do they mean? Photosensitizers are specific agents applied for photodynamic treatment rather than staining organelles.
- the paper needs substantial language correction. Some of the examples are: “more fluorogenic can be transferred…”, “spectra did not obvious change “, “exhibiting green in an AIE-active manner”, “absorption of UV-vis spectrum”, “should result from be caused”, “QC of UV−vis absorption spectra”, “The MTT outcome showed that the IOSE-80 and L929.”, “Viability of relative cell L929 cells”, “the samples were incubated with IOSE80 cells”, etc. etc. etc.
Round 2
Reviewer 2 Report (Previous Reviewer 2)
I could see that all questions/suggestions were appropriately answered/addressed by the authors. In this way, I recommend the manuscript publication
Author Response
We want to express our sincere gratitude for your assistance and guidance throughout this process. Your contributions have been invaluable, and we are truly grateful for your support.
Reviewer 3 Report (New Reviewer)
I think this manu can be published. Congratulations!
Author Response
We appreciate the time and effort you dedicated to thoroughly reviewing our work. We sincerely thank you for your valuable feedback that we have used to improve the quality of our manuscript.
Reviewer 4 Report (New Reviewer)
The authors noticeably improved the paper following the Reviewers’ concerns. However there are still several unresolved issues that do not allow me recommending the paper for publication. These are:
1. Authors did not give any idea why fluorescent properties of their dye were tested in a number of non-biological media. Please explain. And after my comment they added some data to SI instead of adding them to the paper. The absorbance/fluorescence properties of both QC and AC-QC in PBS should be moved to the paper, being added to Fig. 2 or shown in a separate figure.
2. I am still not convinced by the colocalization images in Fig. 5. Authors presented high-res images for cells with AC-QC, but images with organelle trackers are of too low resolution. They need to be replaced and therefore merged images and the analysis also need to be updated. In the current form only data with nuclei tracker seems convincing.
When correcting the figure, please also indicate, at least in the caption, what is depicted in each column. It is obvious for authors but may not be obvious for a reader.
Also, the PC values shown in the figure differ from those given in the text.
3. Authors did not respond to my question on power densities of the radiation used in experiments on photostability of Mitotracker and their dye. If these power densities were quite different then the results cannot be compared.
4. A number of figures shown in the SI are not referred in the text and have very concise captions. What for they are presented?
5. The paper still contains a huge number of language errors, especially in the newly added parts. I understand that the authors are not native speakers, but please do correct. Currently reading is quite frustrating.
Author Response
We appreciate the time and effort you dedicated to thoroughly reviewing our work, and we are grateful for the thoughtful and constructive feedback you provided. Your comments have been instrumental in refining our ideas and enhancing the clarity of our writing. Once again, We want to express our sincere gratitude for your assistance and guidance throughout this process. Your contributions have been invaluable, and I am truly grateful for your support. Please see the point-by-point response to the comments in the attachment.

This manuscript is a resubmission of an earlier submission. The following is a list of the peer review reports and author responses from that submission.
Round 1
Reviewer 1 Report
The manuscript “Mitochondria–Targeted Fluorescent Nanoparticles with Large Stokes Shift for Real-Time Imaging” by Li Xiao and all. Describes the synthesis an fluorescent properties of Ac-Qc pseudo peptide. The synthesis of the QC has been checked by NMR and the AC-QC probe by confocal microscopy. In general the topic is interesting, however in its present state the manuscript is not suitable for publication in Molecules.
Major concerns.
Don’t really understand the abstract construction: (1) Background, (2) … (4) conclusions. Please remove the 1, 2 , 3 and 4. Actually the suggestion is to rewrite the abstract, in its present form it is more like an enumeration.
I am not sure if MITO is a biomarker by itself. Normally biomarkers target something in the mitochondria or are related to mitochondrial enzymes …. like the AC-QC probe.
1. The authors have not shown that the synthesized marker is related to some “disease” e.g. cancer ADs etc.. Basically it stains MITO, but it is not specific so this is a concern.
3. Note a real concern but a MITO tracker red/green price is around 230 USD (+ other dyes). The price alone of the used 6 mg peptide is actually higher + the several steps of the QC reaction. Unless the AC-QC is really targeting something specific it is a nice scientific work but the design/concept must be rechecked.
https://www.cellsignal.com/products/buffers-dyes/mitotracker-green-fm/9074
https://www.cellsignal.com/products/buffers-dyes/mitotracker-red-cmxros/9082
3. The characterization of AC-QC “peptide” is really scarce. What is the AC-QC yield? How are you sure that the AC-QC product is pure? The use of amicon type concentrator is not a guarantee for purity. Additional HPLC HPLC/MS against the starting peptide alone must be undertaken (or other method).
4. How do you know(determine) the used AC-QC concentrations in the 2.3. In vitro cytotoxicity?
5. The sequence in the text and in the Si differ: in the text 4 r-Cha in the SI 3 r-Cha?
Ac-Lys-(D-Arg)-(Cha)-(D-Arg)-(Cha)-(D-Arg)-(Cha)-(D-Arg)
AC-Kr-(Cha)-r-(Cha)-r-(Cha)-r) (Remark:r is D-Arg) ; Guess it is K-r-
3 r-Cha in scheme 3 .
Could you provide the link to the peptide in Sigma/Merck most are TFA salts.
Minor
Addition data about the data processing is required, software etc. especially for confocal microscopy.
Please check carefully mitochondrion, singular vs Mitochondria L16 e.g. Mitochondria plays or play? The authors tend to mix plural and singular e.g. fluorescence methods are widely used by virtue of its simplicity (method/methods … its/their simplicity), the ACQ problem hampers …L51
AIE abbreviation is introduced in the abstract but aggregation-induced emission is missing.
L63 AIE luminous? Please explain.
L77 Please provide a reference for the mitochondria targeted peptide.
Scheme 1 is actually a figure and is a conceptualization for AC-QC
Figure 1A. please assign the IR bands to the synthesized QC. The figure resolution is too low to check by zooming.
FIGURE 1B. Is it excitation/emission or it is like 1D absorption? What is the excitation for the measured fluorescence?
FIGURE 1C. Are those ratios or % as in the figure?
L111 Please add respectively: size and morphology by DLS and TEM … respectively
Please provide a supplier for the IOSE and L929 cell lines.
Could you check the yields?
L289 4.3 Absorption Spectra “QC (1 mL, 0.01 mM) was added to 4mL cuvettes” !! 1ml in 4 ml cuvettes, please correct.
Regarding the fluorescence, have you considered studding the effect the solvent e.g. using different solvents may give different Stockes shifts?
In the SI NMR are stated in dmso-d6 whereas in the text chloroform? Please precise
Again in the supporting information KBr pellets not holder.
SI: Exactly what MW are/is determined by LC/MS system.?
Looking at the 1H NMR in the SI the purity of QC is questionable. MS similarly is not pretty.
In conclusions “monitor the viscosity”. The viscosity is not discussed in the text?
References: What is the [J] for? Up to reference 17
Please check the references some journals are abbreviated others not.
Reviewer 2 Report
The manuscript “Mitochondria–Targeted Fluorescent Nanoparticles with Large Stokes Shift for Real-Time Imaging” by Huirong et al. presents the synthesis of NPs for optical sensing of MITO. This investigation is relevant and concerning the Molecules policy, I should indicate this manuscript for publication. Nevertheless, I find that the impact of the work could considerably be augmented by some small changes/suggestions in the manuscript:
- The abstract must highlight the main results of the investigation. In this sense, I suggest (not mandatory) revising. The way that the abstract was prepared sounds strange to me.
- Why did the authors present the results and discussion separately? I could observetha almost 100% of the manuscripts published in Molecules avoid this format. I suggest presenting as one single item "Results and discussion" and then discussing the synthesis and the respectrive spectroscopic characterizatinon and then presenting the photophysical characterization.
Figure 1 is consufing:
- Figure 1A can be removed since the author does not even mention the results from this analysis.
- Figure 1D, why only absorbance was perfored in DMSO? Why the curve was presented only u to 450 nm? If the curve is normalized, please remove "(a.u.)"
- Scheme 2: all chemical structures must be numbered. Please present the synthesis conditions in the respective arrows or in the Scheme's caption. (the same as Scheme 3)
- Why 13C NMR data was not presented?
Reviewer 3 Report
In this manuscript, the authors sought to produce nanoparticles derived from a conjugate between a quinoxalinone scaffold and a mitochondria-target peptide to be used as a AIE-based probe to image mitochondria.
Unfortunately, the paper lacks proper chemical characterization, for which reason I recommend rejection.
For instance, for compounds 1-3 only a list of 1H and 13C chemical shifts is provided, without presenting the respective spectra in SI. For these compounds HRMS is presented but the observed values are well outside the 0.003 m/z limit. Compound 4 is not characterized at all. The last step of the synthesis of QC is not even described, although 1H and 13C NMR spectra are provided in the SI and clearly show that the QC product is not chemically pure. Finally, the synthesis of conjugate AC-QC is described but no characterization is provided at all.
Round 2
Reviewer 1 Report
The revised version of the manuscript “Mitochondria–Targeted Fluorescent Nanoparticles with Large Stokes Shift for Real-Time Imaging” by Li Xiao et al. has been presented. The abstract has been rewritten, the remaining parts of the manuscript are basically kept.
On figure 1B the blue (see legend) is the examination of absorption with a peak at 400 nm. On 1C the same absorption is shown to peak at 500 nm? Then again figure 1abcd? Which one is it?
The responses to point 3 and 4 are unclear to me. By weighing the AC-QC or something else?
Please state clearly 3.3 Synthetic route of AC-QC nanoparticles how may mg of Ac-QC did you obtain after purification. Was it in solution or powder form?
The provided LC is for AC alone . Is it possible to inject AC and AC-QC together and get a clear separation for the two compounds?
The QC alone should provide a peak around 445. The HRMS spectra of QC show 468-469? Is it Na derivatized?
The comparison of AC and AC-QC MS is a bit problematic – the results look identical to me. Both show signals at 425 and 637 m//z. Please explain the assignment as the attachment of a 445 MW should clearly affect the signals.
That’s all
Reviewer 3 Report
I maintain my recommendation to reject the paper.
It is clear from the 1H NMR spectra presented in Figure 1 from the supporting information that compounds 7, 8 and 9 are not pure. It is also clear that the QC compound (a fundamental component of the AC-QC nanoparticles) is not pure, judging from the provided spectra in figs 3 and 4 in the supporting information, namely the 1H, 13C NMR and HRMS (QS-H+ has a calculated mass of 446.4825, while the HRMS shows a peak of 468.1532).
Most importantly, Figure 5 in the supporting information shows the LC/MS spectra of AC-QC where the two peaks observed actually correspond to the unconjugated peptide AC (compare with MS spectrum in Fig 7). This means that the conjugation between QC and AC failed, which means there are no AC-QC nanoparticles.
